# Structural elucidation of a novel mechanism for the bacteriophage-based inhibition of the RNA degradosome

An Van den Bossche[1,2*†], Steven W Hardwick[3†], Pieter-Jan Ceyssens[1,2], Hanne Hendrix[1], Marleen Voet[1], Tom Dendooven[1], Katarzyna J Bandyra[3], Marc De Maeyer[4], Abram Aertsen[5], Jean-Paul Noben[6,7], Ben F Luisi[3*‡], Rob Lavigne[1*‡]

[1]Laboratory of Gene Technology, KU Leuven, Leuven, Belgium; [2]Division of Bacterial diseases, Scientific Institute of Public Health, Brussels, Belgium; [3]Department of Biochemistry, University of Cambridge, Cambridge, United Kingdom; [4]Biochemistry, Molecular and Structural Biology Scetion, KU Leuven, Leuven, Belgium; [5]Laboratory of Food Microbiology, KU Leuven, Leuven, Belgium; [6]Biomedical Research Institute, University of Hasselt, Diepenbeek, Belgium; [7]Transnational University Limburg, University of Hasselt, Diepenbeek, Belgium

*For correspondence: An. vandenbossche@wiv-isp.be (AVdB); bfl20@cam.ac.uk (BFL); rob.lavigne@biw.kuleuven.be (RL)

†These authors contributed equally to this work
‡These authors also contributed equally to this work

Competing interests: The authors declare that no competing interests exist.

**Abstract** In all domains of life, the catalysed degradation of RNA facilitates rapid adaptation to changing environmental conditions, while destruction of foreign RNA is an important mechanism to prevent host infection. We have identified a virus-encoded protein termed gp37/Dip, which directly binds and inhibits the RNA degradation machinery of its bacterial host. Encoded by giant phage φKZ, this protein associates with two RNA binding sites of the RNase E component of the *Pseudomonas aeruginosa* RNA degradosome, occluding them from substrates and resulting in effective inhibition of RNA degradation and processing. The 2.2 Å crystal structure reveals that this novel homo-dimeric protein has no identifiable structural homologues. Our biochemical data indicate that acidic patches on the convex outer surface bind RNase E. Through the activity of Dip, φKZ has evolved a unique mechanism to down regulate a key metabolic process of its host to allow accumulation of viral RNA in infected cells.

## Introduction

The process of RNA turnover following transcription is vital in the regulation and quality control of gene expression. In γ-proteobacteria, several key constituents of the mRNA decay machinery are assembled in the membrane-associated RNA degradosome (*Aït-Bara et al., 2015*; *Marcaida et al., 2006*). In the paradigm *Escherichia coli* RNA degradosome, this complex is built around the hydrolytic endoribonuclease RNase E, which initiates the rate-limiting step in RNA degradation (*Del Campo et al., 2015*; *McDowall et al., 1995*). Subsequent degradation is carried out by the 3'-5' phosphorolytic exoribonuclease PNPase (polynucleotide phosphorylase) assisted by the ATP-dependent helicase RhlB (both of which are components of the degradosome assembly) and is completed by an oligo-ribonuclease which is not associated to the complex (*Evguenieva-Hackenberg and Klug, 2011*; *Górna et al., 2012*). The protein composition of the RNA degradosome varies among proteobacteria and during various stages of growth (*Carabetta et al., 2010*; *Ikeda et al., 2011*).

RNase E, a member of the RNase E/G family, is a tetrameric enzyme and can be broadly divided into two functional halves. The N-terminal half (NTH) comprises the catalytic domain, while the non-conserved C-terminal half (CTH) is natively unstructured and acts as a scaffold to assemble the complex (*Aït-Bara et al., 2015*; *Callaghan et al., 2005*). Despite the predicted lack of structure within the scaffold domain, several short segments having structural propensity were identified in the *E. coli* CTH that mediate the interaction between RNase E and the cell membrane, enolase and PNPase. Moreover, the CTH contains two arginine-rich regions that have been shown to bind RNA: the RNA binding domain (RBD)/Arginine-rich region 1 (AR1) and AR2 (*Callaghan et al., 2004*). The activity and specificity of the RNA degradosome is under complex regulation and is influenced by several factors including riboregulation by small, non-coding RNAs (sRNA) and proteins RraA and RraB ('Regulators of RNase Activity A and B') that inhibit the activity of RNase E by binding to the CTH (*Górna et al., 2010*; *Ikeda et al., 2011*; *Zhou et al., 2009*).

During infection, lytic bacteriophages create a favourable environment for the generation of progeny by influencing the activity and specificity of host proteins (*Roucourt and Lavigne, 2009*). Three cases have been reported in which the machinery of RNA decay is a target of phage effector proteins. In one, *E. coli* phage T7 heavily phosphorylates the CTH of RNase E and RhlB, leading to the inhibition of RNA degradation (*Marchand et al., 2001*). In a second example, the Srd protein encoded by *E. coli* phage T4 was found to increase the activity of RNase E on host mRNA by binding to the catalytic NTH (*Qi et al., 2015*). This may account for earlier observations that, during infection by phage T4, the host mRNA was destabilized while the phage mRNA was stabilized (*Ueno and Yonesaki, 2004*). Finally, an increase in the expression of RNase E was observed during the infection of *Prochlorococcus* MED4 by cyanophage P-SSP7, due to elevated levels of an RNase E mRNA variant lacking the 5'UTR responsible for the negative feedback regulation of the gene. In parallel, antisense RNAs derived from the phage sequester the P-SSP7 transcriptome to form dsRNA, which is subsequently protected from degradation by RNase E (*Sesto et al., 2013*; *Stazic et al., 2016*, *2011*).

Being one of seven known genera of lytic phages infecting *Pseudomonas aeruginosa*, giant 'φKZ-like' bacteriophages form a remote branch of myoviruses. The φKZ virus type possesses an unusually large 280 kb genome with little evolutionary relation to other known genera. Its genomic G/C content is remarkably low (36.3%) in comparison to its host *P. aeruginosa*, implying a short history of co-evolution (*Mesyanzhinov et al., 2002*). Notably, φKZ infection causes more than five-fold increase in total cellular RNA, suggesting that the phage affects *P. aeruginosa* RNA biogenesis and breakdown (*Ceyssens et al., 2014*).

In an effort to understand the φKZ infection process and its impact on RNA metabolism, we identified potential interaction targets of viral proteins in the opportunistic pathogen *P. aeruginosa* (*Ceyssens et al., 2014*). By performing a screen based on affinity purification and mass spectrometry, we observed that a previously uncharacterised phage protein specifically binds to two RNA binding sites within the CTH of RNase E, and by doing so efficiently inhibits the RNA binding and degradation activity of the degradosome assembly. We have determined the structure of this protein (termed Dip, 'degradosome interacting protein') by X-ray crystallography and show that it forms an open-clamp like homo-dimeric structure, and binding of Dip to RNA binding regions within RNase E is characterized functionally and structurally. To our knowledge, Dip is the first known viral protein which effectively inhibits the RNA degradation activity of its host via a direct protein:protein interaction.

## Results

### A φKZ protein co-purifies with the RNA degradosome

To identify potential phage proteins that interact with the host RNA degradation machinery, a pull down experiment was designed in which a *Strep*-tag II was fused on the C-terminus of RNase E of *P. aeruginosa* strain PA01. The modified strain was subsequently infected with a collection of seven different *P. aeruginosa*-specific phages, and a phage protein was co-precipitated only in cells infected with giant phage φKZ (*Supplementary file 1-Table 1*). This Degradosome interacting protein (gp37 – referred to hereafter as Dip) has a predicted molecular weight of 31.7 kDa and has no sequence similarity with any protein in currently available databases.

A reciprocal in vitro pull down assay was performed in which *P. aeruginosa* cell lysate was applied to Dip immobilised on a Ni²⁺ affinity column through a hexa-histidine-tag. Compared to the control reactions in which only Dip or cell lysate was used, several protein bands were clearly enriched (*Figure 1A*). The most abundant co-purifying band was identified as RNase E by mass spectrometry analysis, and other predicted components of the RNA degradosome (PNPase and DeaD, which were

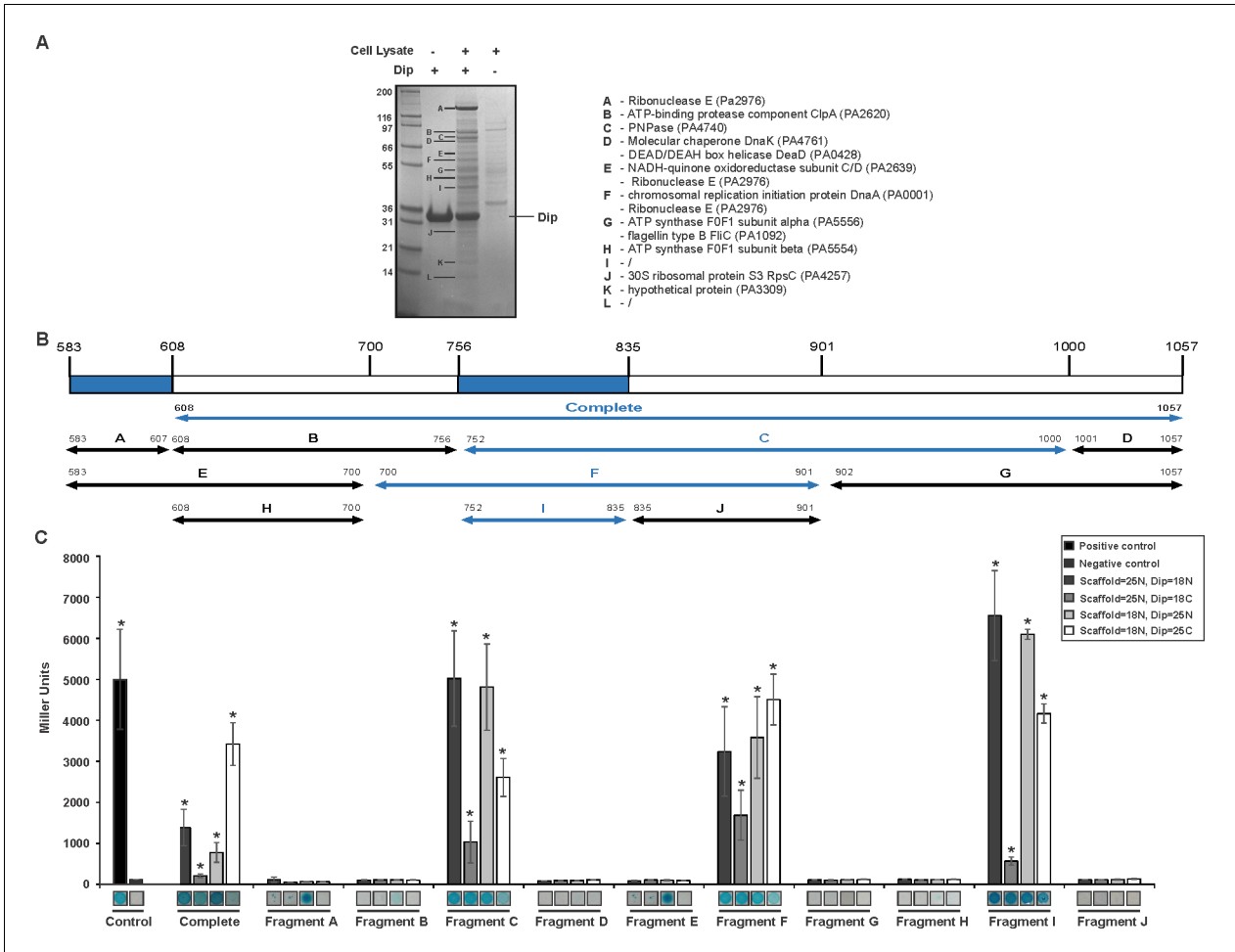

**Figure 1.** Interaction analyses of Dip and the *P. aeruginosa* RNA degradosome. (**A**) In vitro pull down of *P. aeruginosa* cell lysate, using his-tagged Dip as a bait. Eluted samples were loaded on a 12% SDS-PAGE gel. The letters indicate bands submitted for identification by mass spectrometry analysis ("/" represents bands which could not be confidently identified). (**B**) Fragments of the scaffold domain of RNase E used in the bacterial two-hybrid assay. Blue and black arrows indicate fragments with a positive and no signal, respectively. The numbers indicate the number of the residue of RNase E. (**C**) Bacterial two-hybrid assay in which the T25 (25) or T18 (18) domain of CyaA is fused to the N-terminal (N) or C-terminal (C) side of a target protein. Non-fused T25 or T18 domains were used as negative control (No insert). The leucine zipper of GCN4 was used as a positive control. Interactions were visualized by a drop test on selective medium (shown below the graphs) and β-galactosidase activity was measured quantitatively in Miller units. Error bars represent SD and *P*-values were calculated using Student's t-test (n = 3), *p<0.05.

The following source data and figure supplements are available for figure 1:

**Source data 1.** Data bacterial two-hybrid.

**Figure supplement 1.** ELISA using 350 nM of Dip (fused to a C-terminal his-tag) as a bait protein and increasing amounts of *P. aeruginosa* RNA degradosome (carrying a C-terminal Strep-tag II on RNase E) as a prey (numbers in nM).

**Figure supplement 1—source data 1.** Data ELISA.

also identified in the original pull down, *Supplementary file 1-Table 1*) were among the co-purifying proteins.

To corroborate this interaction in vitro, an ELISA was performed using Dip as bait and the *P. aeruginosa* RNA degradosome as prey. Titrating a constant amount of Dip with RNA degradosome yielded an increasing signal which saturated at an approximate ratio of 10 Dip protomers for each degradosome protomer (*Figure 1—figure supplement 1*). Together, these results confirm that the primary binding partner of Dip in the *P. aeruginosa* host is the RNA degradosome assembly.

## Dip binds to specific regions of RNase E

A bacterial two-hybrid assay was designed to determine which component(s) of the RNA degradosome are targeted by Dip. Initially PNPase, three DEAD-box helicases and RNase E (divided into NTH and CTH) were assessed for their interaction with Dip, and a significant ($p < 0.05$) positive reaction with Dip was only observed for the C-terminal part of RNase E (CTH; residues 608–1057) (*Figure 1B–C*).

A second bacterial two-hybrid assay was designed to more precisely map the interaction site within the C-terminal half of RNase E. The CTH was divided into ten overlapping fragments based on homology to molecular recognition sites within RNase E of *E. coli* and predictions of protein-protein binding sites by the online tool ANCHOR (*Dosztányi et al., 2009*) (*Figure 1B*). Two fragments (delineating the region 583–607 of RNase E) yielded a positive signal on selective medium for one vector combination (*Figure 2A*, fragments A and E). However, this signal could not be confirmed using Miller assays (*Figure 2B*). For three other fragments (all-encompassing residues 756–835 of RNase E, *Figure 1B* fragments C, F and I), a strong signal was produced for all vector combinations and could be positively confirmed in subsequent Miller assays ($p < 0.05$) (*Figures 1B–C*, fragments C, F and I). Combining these results, a clear interaction site can be defined within residues 756 to 835, while a weak second interaction site may be present within residues 583–607. These binding sites were subsequently verified by electrophoretic mobility shift assays (EMSA). *Figure 2A* demonstrates that the mobility of Dip is altered upon addition of increasing concentrations of 583–607 (fused to a GST-tag), with all of the Dip being shifted at an approximate 10-fold excess of the RNase E fragment. When using a slightly larger fragment of RNase E (residues 583–636), Dip is shifted completely by only a four-fold excess of the GST-fusion protein. This suggests that efficient Dip binding requires a larger region of RNase E than the first defined segment of residues 583–607. This might also explain why the bacterial two-hybrid assay with residues 583–607 and Dip was inconclusive.

Bioinformatic analyses of the second Dip binding fragment of RNase E (residues 756–835) suggested that this region may encompass an RNA binding motif at residues 757–772 (BindN; *Wang and Brown, 2006*) and a possible protein binding site at residues 776–835 (ANCHOR). In *Figure 2B*, only a clear shift of Dip is visible upon addition of the RNase E fragments harbouring the predicted RNA binding site of residues 757–772 (756–901, 756–835 and 756–775). Dip is efficiently shifted at just a 1:1 ratio to the different fragments, suggesting stronger binding to residues 756–775 than 583–636 of RNase E. To ensure that the binding is not mediated by the presence of RNA, an additional EMSA was performed using 9S rRNA and increasing amounts of Dip. Since no shift of the RNA was observed (*Figure 2—figure supplement 1*), we conclude that Dip is not an RNA binding protein. Therefore, the interaction between Dip and the 583–636 and 756–775 regions of RNase E of *P. aeruginosa* is via a direct protein-protein interaction.

## The binding sites of Dip are conserved in *E. coli*

When comparing both Dip interaction sites (583–636 and 756–775) found in *P. aeruginosa* RNase E to the RNase E of *E. coli*, these arginine rich regions were found to align well to the known RNA binding sites RBD and AR2, respectively (data not shown). These findings suggest that the binding sites of Dip may be conserved in RNase E of other bacteria. To investigate whether Dip is capable of binding to the RNA degradosome of other species, an in vitro pull down was performed using his-tagged Dip and *E. coli* cell lysate (*Figure 3A*). The RNA degradosome components RNase E and PNPase, and the ribosomal protein L13 were identified as the predominant bands identified by mass spectrometry, while two other canonical degradosome proteins, RhlB and enolase, were also identified. An EMSA using purified recombinant proteins corresponding to the CTH of *E. coli* RNase E equally showed a clear shift of Dip in the presence of the CTH at the 1:1 molar ratio (*Figure 3—*

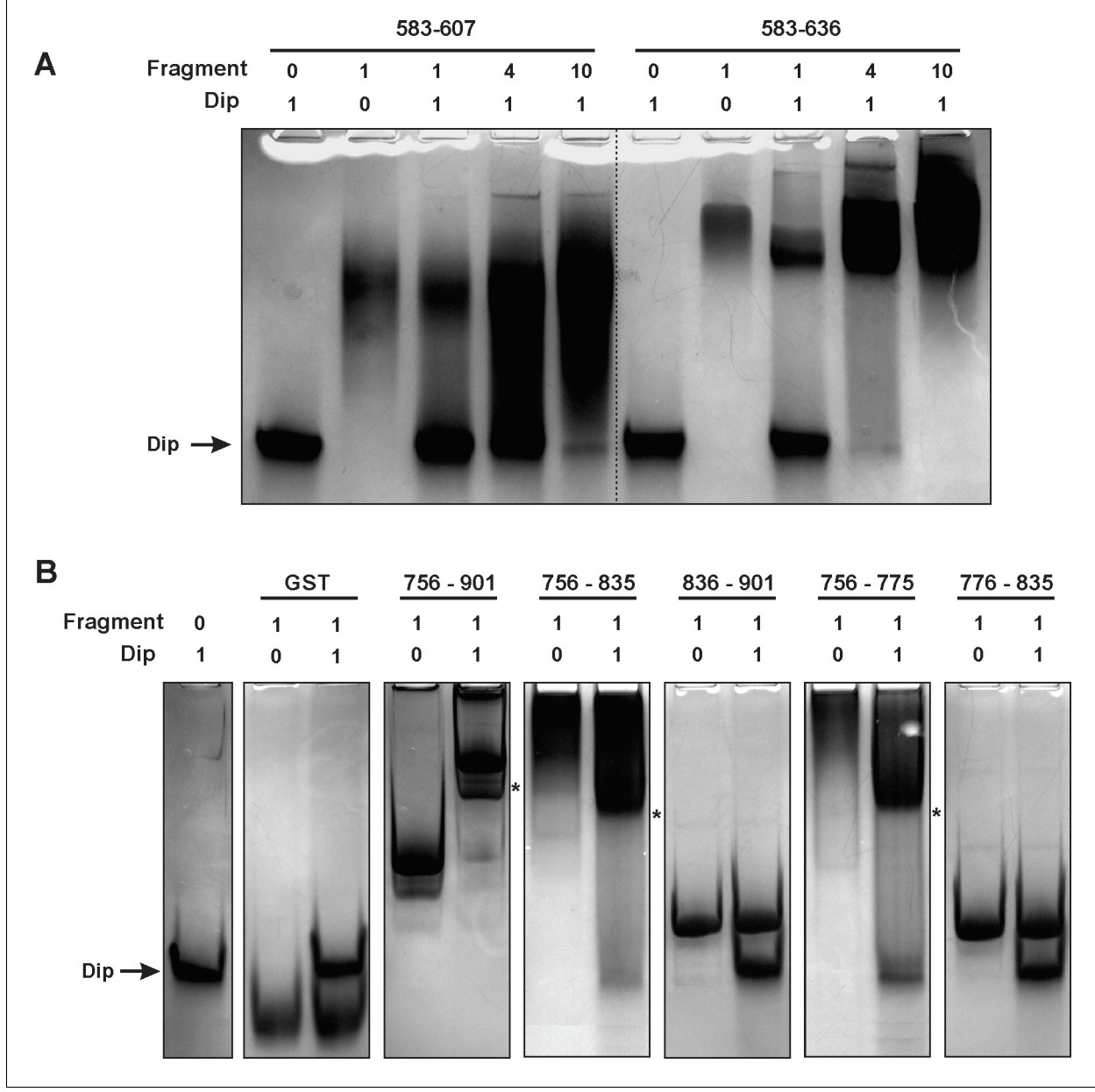

**Figure 2.** Electrophoretic Mobility shift assays using Dip. (**A**) EMSA of fragments of RNase E corresponding to a first interaction site for Dip, incubated with or without Dip. The numbers above the horizontal line indicate the residues of RNase E corresponding to the two tested fragments. The numbers below indicate the relative amount of the fragments and Dip. (**B**) EMSA using fragments of the RNase E belonging to the second site of interaction for Dip. The numbers indicate the residues of RNase E that encompass the fragments. A shift in migration is indicated with an asterisk.

The following figure supplement is available for figure 2:

**Figure supplement 1.** EMSA of increasing amounts of Dip (0.5 pmol, 1 pmol, 5 pmol, 10 pmol and 40 pmol) and 1 pmol of 9S RNA of *E. coli*.

*figure supplement 1*). This demonstrates that Dip forms a direct protein-protein association with *E. coli* RNase E, which is not dependent upon other proteins or RNA.

## Dip competes with RNA for binding to RNase E

To test the hypothesis that Dip targets RNA binding sites within RNase E, an EMSA in the presence of RNA was performed. The 756–901 (GST-tagged) and 756–775 (untagged) fragments of *P.*

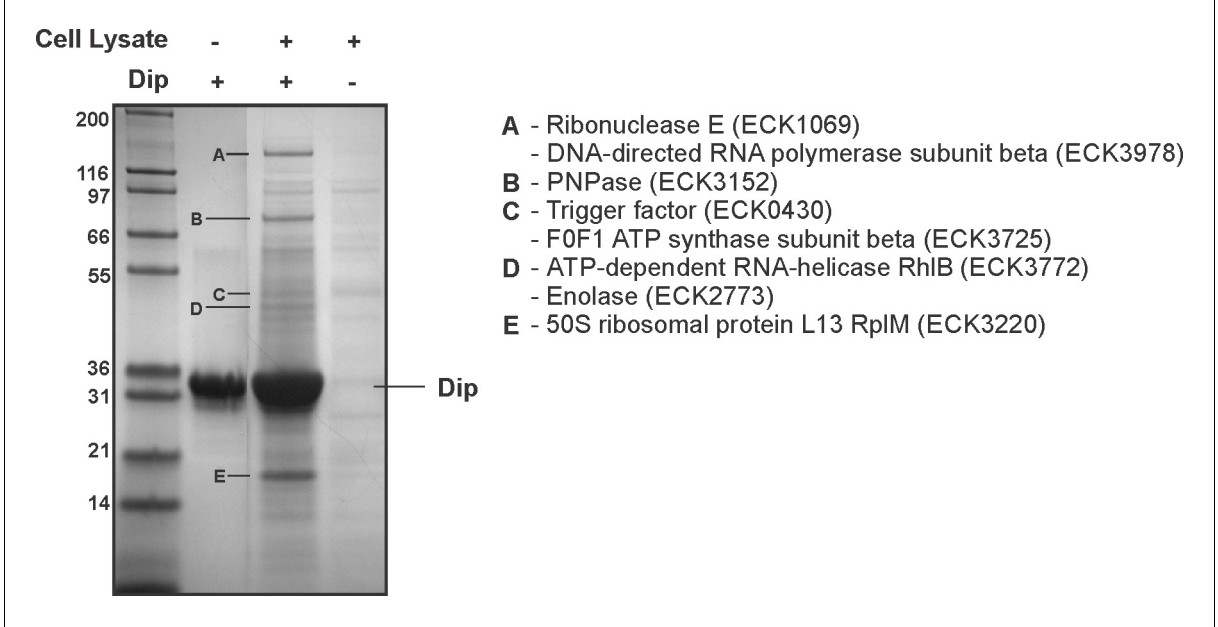

**Figure 3.** Interaction analyses of Dip and the *E. coli* RNA degradosome. In vitro pull down of *E. coli* cell lysate, using his-tagged Dip as a bait. Eluted samples were loaded on a 12% SDS-PAGE gel. The letters indicate proteins identified by mass spectrometry analysis.

The following figure supplement is available for figure 3:

**Figure supplement 1.** EMSAs of Dip and the CTH (catalytic half (1–26/498–1061) of the *E. coli* RNase E. Sample were run on an 8% native acrylamide gel.

*aeruginosa* RNase E were mixed with a short RNA oligonucleotide (27mer). In the presence of the 756–901 fragment a clear shift of both the protein and the RNA was visible (**Figure 4A**). The untagged 756–775 fragment was not able to enter the gel in isolation; however, in its presence there is a visible shift of the RNA (**Figure 4—figure supplement 1A**). When increasing amounts of Dip were subsequently added to these RNA:RNase E complexes, a new protein:protein complex is formed between Dip and the RNase E fragments, while the bound RNA is released and returns to a position at the bottom of the gel (**Figure 4A**). These observations show that Dip is able to compete with and displace RNA from the RNase E fragments.

The same RNA binding assay was performed in the presence of the 583–636 fragment of *P. aeruginosa* RNase E, but no shift of the protein fragment or the RNA could be visualized when using the 27mer RNA (**Figure 4—figure supplement 1B**). However when using the larger 9S rRNA (245 nt, precursor of 5S rRNA), a clear shift can be observed (**Figure 4B**). As with the 756–775 binding site, 9S rRNA can be competitively displaced from the 583–636 fragment by Dip, and the addition of a 4-fold excess of Dip is able to completely remove the bound RNA from the RNase E fragment. These results demonstrate that Dip targets both RNA binding sites in the C-terminal half of the RNA degradosome, and consequently blocks the binding of RNA to the RNase E subunit.

## Dip inhibits in vitro RNA degradation by the RNA degradosome

Since Dip prevents/displaces RNA from binding to the degradosome, we examined in vitro the functional consequences of this interaction. A well characterised and conserved activity of RNase E is the processing of 9S ribosomal RNA to the precursor p5S (**Cormack and Mackie, 1992**; **Hardwick et al., 2011**). We tested 9S rRNA of *E. coli* as a substrate for in vitro activity assays using both the *P. aeruginosa* and the *E. coli* RNA degradosome in the presence and absence of Dip. First, Dip itself was tested for RNase activity towards the 9S rRNA fragment. After incubating the RNA fragment with the phage protein for 30 min, no RNA degradation was observed (**Figure 5—figure supplement 1A**). Next, the RNA degradosomes were incubated with 9S rRNA in the absence of

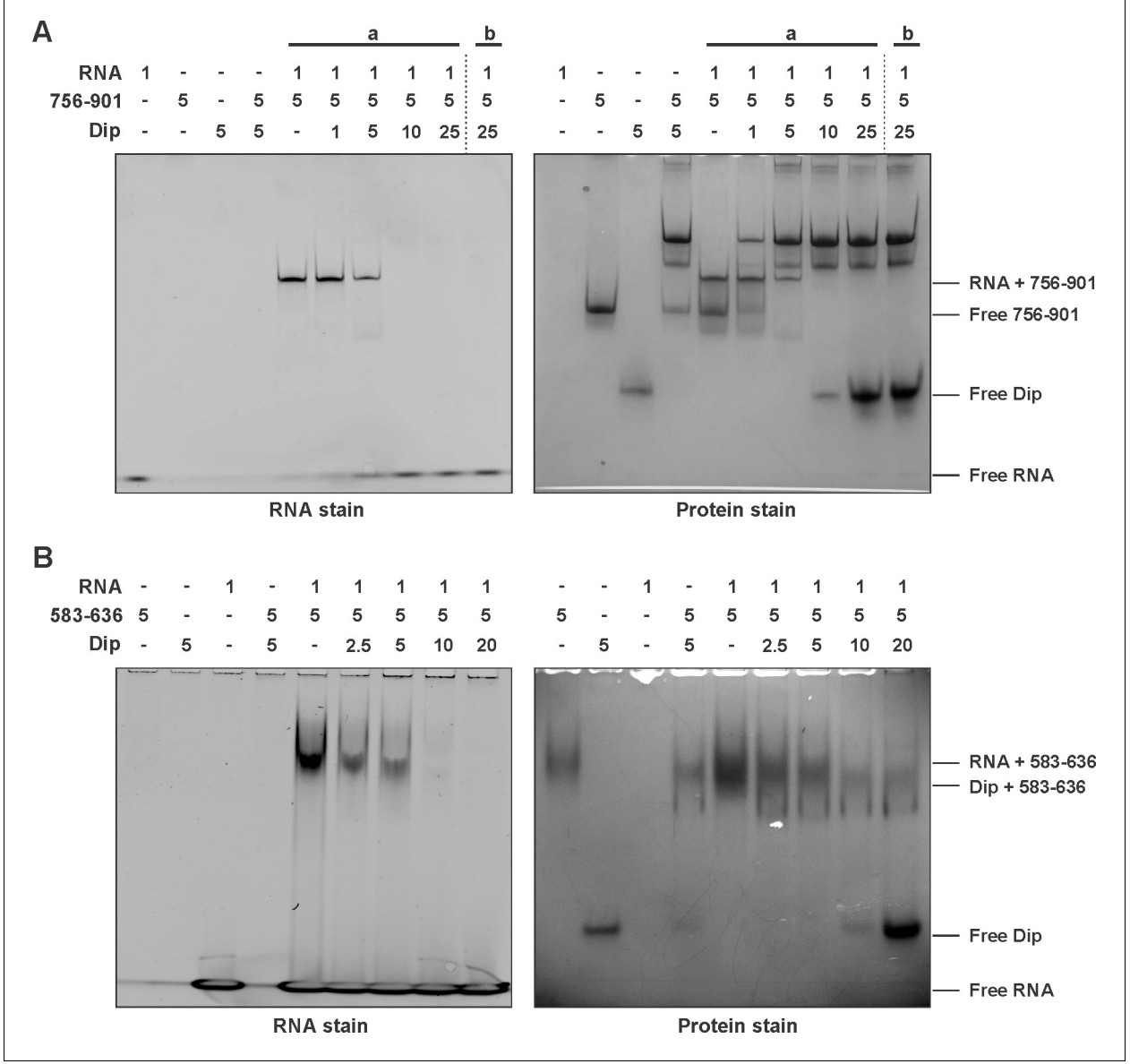

**Figure 4.** Competitive binding of RNA and Dip to RNase E. (**A**) EMSA of RNA (27mer), the 756–901 fragment of RNase E (fused to a GST-tag) and Dip. (a) indicates that RNA was incubated with the fragment prior to the addition of Dip. (b) indicates that Dip was incubated with the peptide before adding the RNA. (**B**) EMSA of 9S RNA, the 583–636 fragment of RNase E (fused to a GST-tag) and Dip. The samples were run on an 8% native acrylamide gel. Concentrations are presented in μM.

The following figure supplement is available for figure 4:

**Figure supplement 1.** Electrophoretic mobility shift assay of RNase E peptides, 27mer RNA and Dip.

Dip, resulting in efficient RNA processing (*Figure 5A*). With increasing amounts of Dip, the processing of 9S rRNA is visibly reduced, with almost no cleavage observed in the presence of a 10-fold access of Dip. Therefore, we conclude that Dip inhibits the processing activity of the RNA degradosome of both *P. aeruginosa* and *E. coli*. To verify if these observations are specifically due to the binding of Dip to the scaffold domain of RNase E, an in vitro degradation assay was performed using only the catalytic domain of *E. coli* RNase E (residues 1–525). We noted that RNase E was still capable of processing 9S RNA (albeit less efficiently than the full degradosome), but no inhibition of the

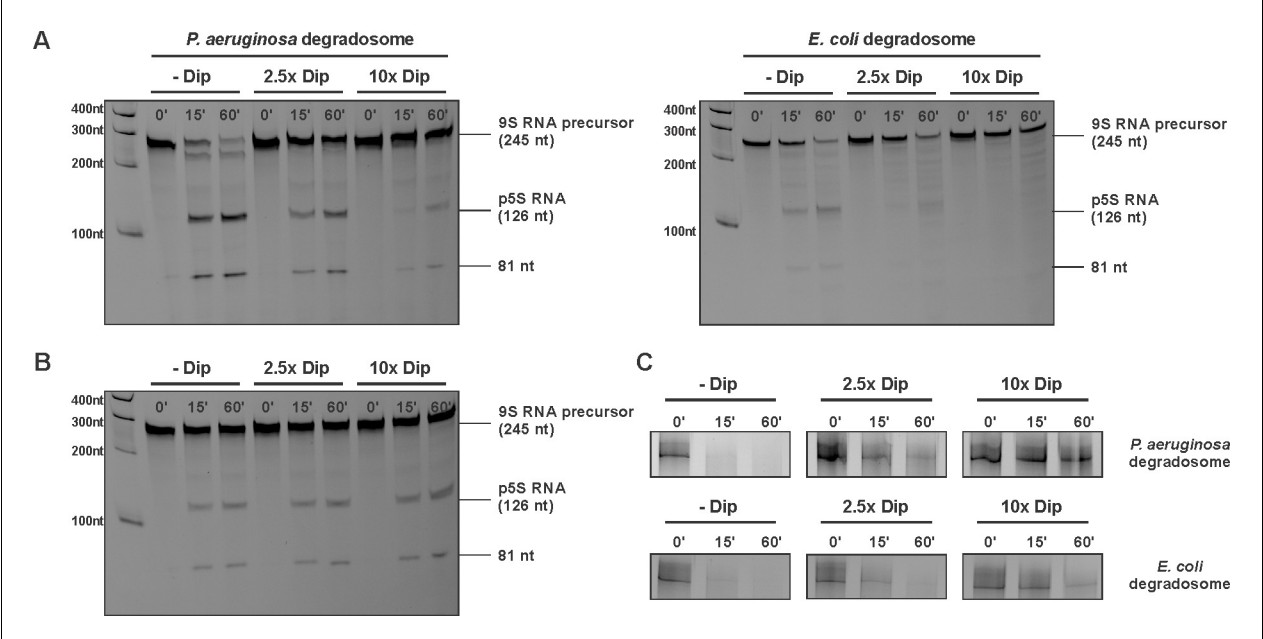

**Figure 5.** Degradation and processing assays in the presence of Dip. (A) The 9S RNA precursor of 5S rRNA of *E. coli* was incubated in vitro with RNA degradosome in the absence or presence of Dip. (B) *E. coli* 9S RNA was incubated with the catalytic domain of the *E. coli* RNA degradosome (1–529) in the absence or presence of Dip. (C) A late φKZ RNA transcript was incubated with the degradosome in the absence or presence of Dip. All samples were loaded on a 5-8-20% (according to the size) denaturing polyacrylamide gel and visualized with SYBR gold stain.

The following figure supplement is available for figure 5:

**Figure supplement 1.** Influence of Dip on the RNA degradosome activity.

degradation activity could be observed in the presence of Dip (*Figure 5B*), demonstrating that Dip does not directly influence the activity of the catalytic domain.

Additional RNA substrates were assayed to establish whether the inhibitory effect of Dip varies with differing RNA species. When using RNA originating from the φKZ genome, Dip showed a similar inhibitory effect on RNA cleavage as seen with bacterial 9S rRNA (*Figure 5C*). Moreover, the cleavage of the *fdhE* transcript, which has been shown to be processed primarily via a second, 'direct entry' pathway of RNase E (*Clarke et al., 2014*), was efficiently inhibited in the presence of Dip (*Figure 5—figure supplement 1B*). These results indicate that Dip inhibits the activity of the RNA degradosome in vitro without a preference for a specific substrate.

## Dip stabilizes RNA in vivo

Since our in vitro assays point to an inhibitory effect of Dip on the activity of the RNA degradosome, the in vivo role of Dip and its global function during phage infection were further investigated.

RNAseq analysis indicated that the phage protein Dip is transcribed in the early phase of infection, and is highly expressed (*Ceyssens et al., 2014*). To verify the timing of the corresponding protein translation, and to assess Dip stability during phage infection, we performed a western blot on φKZ-infected *P. aeruginosa* samples every 3 min during infection (*Figure 6A*). Using anti-Dip antibodies, Dip was first detected 9 min after the start of infection, reaching a peak at 24 min. Although a decrease in Dip levels is visible at approximately 30 min post infection, the quantity of Dip proteins subsequently increases, after which its levels appear to plateau. This indicates that Dip is produced during the early phase of infection and subsequently persists during the whole infection cycle.

To evaluate the effect of Dip on the bacterial cells, *dip* was cloned into the *E. coli* – *P. aeruginosa* shuttle vector pHERD20T under control of a $P_{BAD}$ promoter, which responds to arabinose in a dose-dependent manner (*Qiu et al., 2008*). When expressing Dip at low levels in *P. aeruginosa* (0.1% arabinose), no variation in growth efficiency could be observed (*Figure 6—figure supplement 1A–*

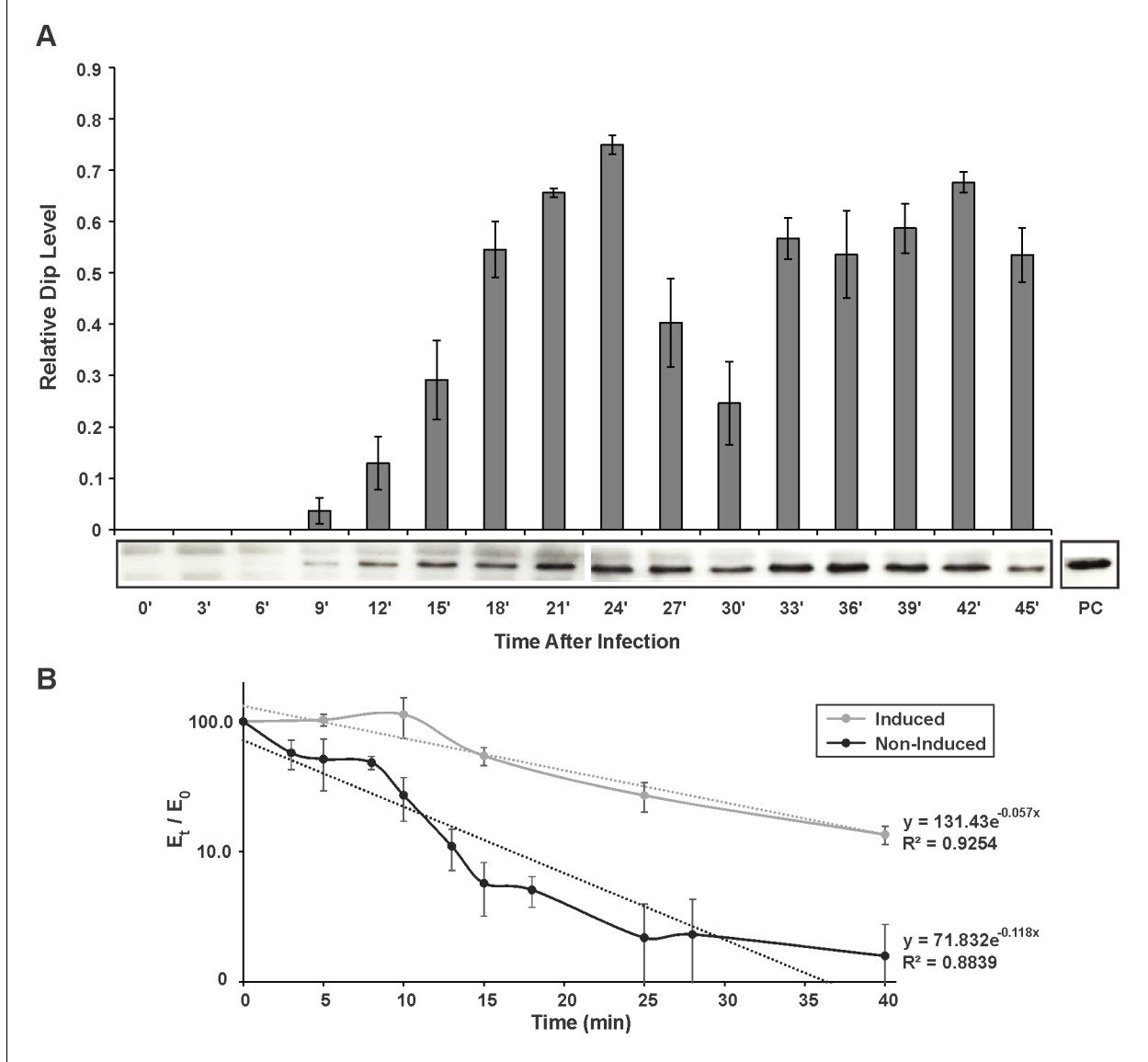

**Figure 6.** In vivo detection and influence of Dip. (**A**) Western blot for the in vivo detection of Dip during φKZ infection using anti-Dip antibodies. Pixel number and intensity were quantified and normalized against the positive control (PC) in imageJ. Error bars represent standard deviation (n = 3). (**B**) In vivo RNA decay of the household gene OprL. The quantity of OprL transcripts from $Rif^{200}$-treated *P. aeruginosa* cells (wild type cells (Non-Induced) or cells expressing Dip) was determined by qRT-PCR and normalized to the total RNA content. The amount of RNA ($E_t$) was compared to the amount of RNA at time point 0 ($E_0$) and plotted in a semi-logarithmic plot as a function of time. Error bars represent standard deviation (n = 3). Dotted lines represent a trend line of a data set.

The following source data and figure supplements are available for figure 6:

**Source data 1.** Data Western blot.

**Source data 2.** Data qPCR.

**Figure supplement 1.** Dip was expressed in *P. aeruginosa* cells from a pHERD20T vector with an arabinose-inducible $P_{BAD}$ promoter.

**Figure supplement 1—source data 1.** Data Bioscreen.

*B*). In contrast, a change in morphology was detected when Dip was expressed at high levels (1% arabinose), with cells changing from the wild type rod-shape into a 'curled' phenotype (*Figure 6— figure supplement 1C*).

The same constructs were used to assess the stability of RNA molecules in the presence of Dip in vivo. The RNA decay of the household gene OprL (peptidoglycan associated lipoprotein precursor) was quantified by qRT-PCR. Although the data show a pronounced variance, they clearly demonstrate that the decay of OprL is slowed in the presence of Dip, with the half-life increasing from ± 6 min to ± 12 min (*Figure 6B*). This is in agreement with the inhibitory effect of Dip on the RNA degradosome observed in vitro.

## Dip is a homodimer forming an open clamp like structure

To help understand the mechanism of Dip, the structure of this protein was solved by X-ray crystallography using selenium SAD experimental phasing (*Supplementary file 1*-Table 2). The 2.2 Å structure of Dip reveals a clamp like homodimeric structure, with the dimeric architecture presenting a grooved, concave face resembling a partially opened scroll (*Figure 7A* and *Figure 7—figure supplement 1*). The dimeric form was confirmed to be the predominant species in solution by analytical ultracentrifugation, while self-interaction was showed during a bacterial two-hybrid analyses (Data not shown). A search through the protein structural database could not identify any significant structural homologues of Dip, indicating that its protomers are composed of an unprecedented α+β fold.

## RNase E binds to the outer surface of Dip

To further study the interaction between Dip and RNase E, modelling experiments were performed to predict the binding surface for one of the Dip binding peptides of *P. aeruginosa* RNase E (residues 756–775). Secondary structure prediction of this RNase E peptide in isolation suggested a high propensity towards a helical conformation. Therefore, this helical form of the peptide was used in a docking experiment intended to scan the complete surface of Dip with the DOT docking program (*Roberts et al., 2013*). The best energetically ranked complexes show binding of the peptide to a negatively charged patch on the outer surface of the Dip-dimer (*Figures 7B–C*).

To verify the in-silico docking results Dip was co-crystallized with the 756–775 peptide of *P. aeruginosa* RNase E. Hexagonal crystals of Dip were obtained containing an apparent ring-like hexamer of Dip formed through crystallographic symmetry. Discontinuous density was observed on the surface of Dip in the DOT docking predicted acidic pocket (*Figure 7D*). Processing the X-ray diffraction data in the lower symmetry P1 space group visibly improved the density for the RNase E peptide, and we were able to cautiously model eight amino acids of the peptide into this density. The model of Dip bound to the short RNase E recognition peptide (Dip:RNaseE$_{756-775}$) indicated significant contacts are made between residues Asp 137, Asp138, Glu 214 and Glu 222 of Dip and the RNase E peptide (*Figure 7C–D*). To confirm the importance of these residues in the interaction between the two proteins, several mutants of Dip were generated where the acidic residues on the surface of Dip were substituted for alanine. A double mutation of Glu 214 and Glu 222 to alanine was sufficient to completely abolish the interaction with both Dip binding sites on RNase E when assessed by an electrophoretic mobility shift assay (*Figure 7E*). Moreover, the in vivo expression of the Dip-E214A/ E222A mutant using the pHERD20T vector restored the wild type phenotype compared to the curly morphology obtained by in vivo Dip expression (*Figure 7—figure supplement 2*). Interestingly, electron density for an unknown small molecule ligand can be seen in this region in the structure of apo-Dip, which we have modelled as a single arginine amino acid. The E214A-E222A mutant of Dip was subsequently crystallized in the same P2$_1$ space group as the wild type Dip protein, and amino acid mutations were confirmed by the absence of electron density for glutamate side chains at positions 214 and 222 in the 2.2 angstrom resolution map. There were no other significant alterations to the local or overall structure of Dip (data not shown). These data show that the acidic patch on the outer surface of Dip is crucial for the interaction with RNase E, and both of the RNA binding segments of RNase E engage Dip via this region.

## Discussion

The RNA degradosome (*Figure 8A*) plays a pivotal role in bacterial post-transcriptional gene regulation, ensuring efficient transcript degradation and processing. In *P. aeruginosa* cells infected with

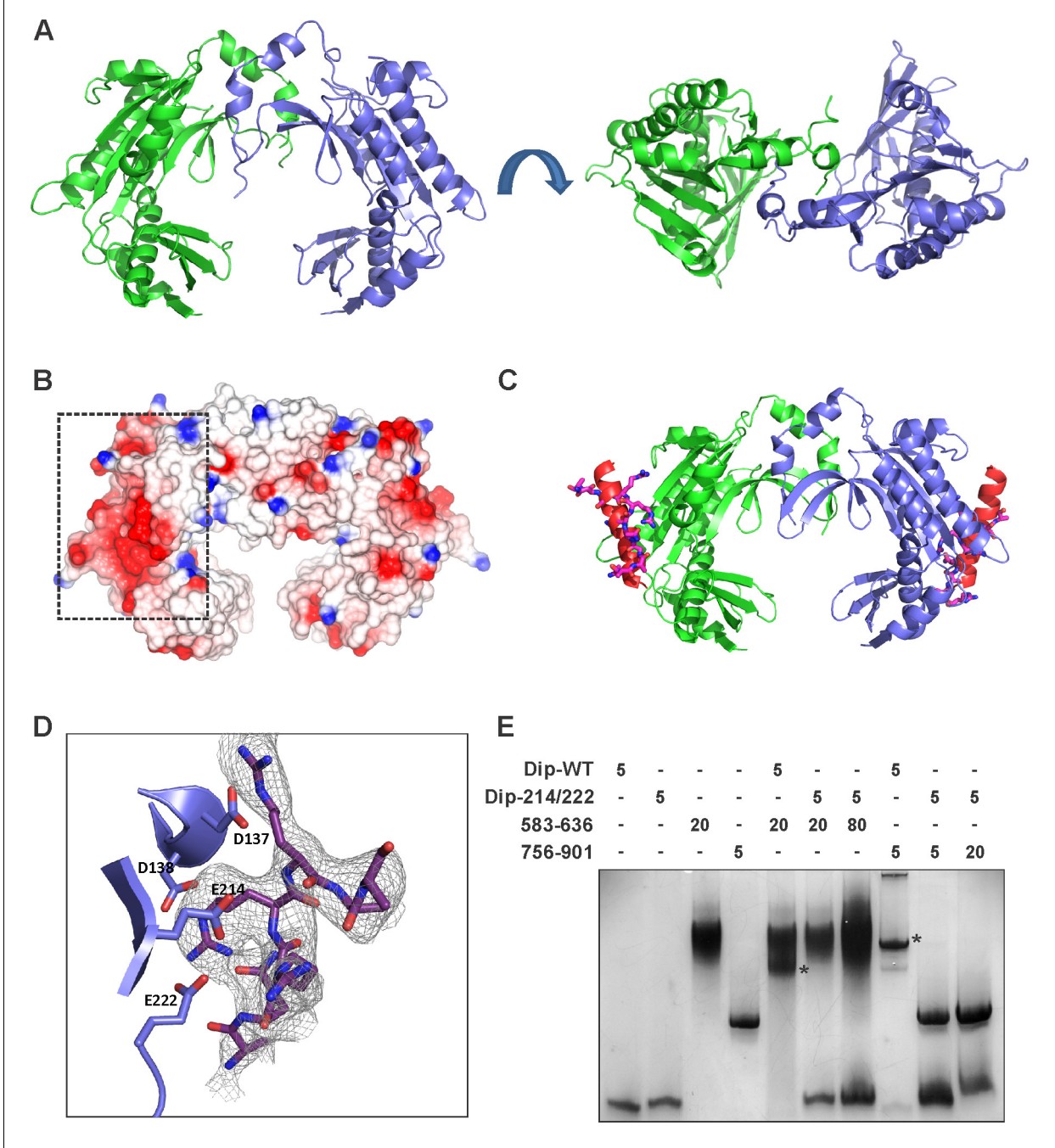

**Figure 7.** The crystal structure of Dip. (A) Two views of the dimeric Dip structure. Protomers are coloured in green and blue. (B) Electrostatic surface representation (red = negative, blue = positive) of the Dip dimer. The negatively charged, patch/pocket on the outer surface of the Dip-dimer is indicated with the dashed box. (C) The structural interaction between peptide 756–775 (of RNase E) and Dip. The in-silico docked peptide is shown as a red helix, and the peptide modelled from X-ray crystal data is shown as purple and blue sticks. (D) Experimental structure of the complex, showing a close view of the interacting amino acids of Dip (D137, D138, E214 and E222) and RNase E peptide 756–775. The electron density map is shown for the RNase E peptide only for clarity. (E) EMSA of wild type Dip and the mutant Dip-E214/E222 (substituted to Ala) and the RNase E peptides 583–636 and 725–901. The samples were run on an 8% native acrylamide gel. Concentrations are presented in µM.

The following figure supplements are available for figure 7:

**Figure supplement 1.** Structural analysis of Dip.

*Figure 7 continued on next page*

*Figure 7 continued*

**Figure supplement 2.** Microscopic view of *P. aeruginosa* cells induced (1% arabinose at $OD_{600\ nm}$ 0.07) for the in vivo expression of No insert (empty vector), Dip and Dip-E214A/E222A using the pHERD20T vector.

giant phage φKZ, we have identified a viral protein 'gp37/Dip' which binds to the putative RNA degradosome assembly. The binding targets of this <u>d</u>egradosome <u>i</u>nteracting <u>p</u>rotein could be clearly defined as two RNA-binding sites within the C-terminus of RNase E (residues 583–636 and 756–775). Our in vitro assays indicate that RNAs bound to these two RNA binding sites are displaced by Dip. Thus, the interaction of Dip with these segments inhibits the activity of the RNA degradosome in vitro and in vivo. To our knowledge, this is the first viral protein identified which uses a direct interaction with the RNA degradosome to inhibit its function. Therefore, Dip can be considered as a viral functional equivalent of the bacterial RraA and RraB ('regulators of RNase activity') proteins of *E. coli*, which inhibit the activity of RNase E by binding to specific regions of the CTH (*Górna et al., 2010*; *Zhou et al., 2009*).

A remarkable observation is that currently no homologues of Dip could be found, not even among φKZ-related phages, raising the intriguing question of how and when the phage acquired this factor. The unique structure of this protein resembles a partially opened scroll or a clamp, and it is tempting to imagine that the two protomers of Dip could act as pincers to capture a binding partner at the dimer interface. However, through a combination of structural, biochemical and bioinformatics analyses we can conclude that the binding site for RNase E is actually on the outer surfaces of the Dip dimer, and not within the "clamp".

Since one dimer of Dip possesses two binding sites on its surface, several possible RNase E binding models can be envisaged. In one scenario, two independent dimers might each bind to a RNA binding site (*Figure 8B*) or alternatively a single dimer may engage both sites at the same time, although the latter would require the two RNA binding segments to be in close proximity (*Figure 8C*). Finally, the possibility that a single Dip dimer may bind to two separate RNase E protomers cannot be dismissed, especially considering that RNase E is a tetrameric enzyme, and the four protomers may be in proximity in a single degradosome assembly (*Figure 8D*).

The question arises as to why the binding of Dip to the RNA-binding sites in the non-catalytic C-terminal region of RNase E should have a significant effect on the RNA processing and degrading activities of the catalytic N-terminal region of the enzyme. Our preferred explanation for these findings is that the RNA binding sites found in the CTH are particularly important for the recognition of RNA substrates that harbour secondary structure elements. Potentially these substrates require unwinding by an RNA helicase bound close to these regions of RNase E prior to cleavage by the NTH of RNase E. Therefore, the binding of structured RNAs to the RNA binding sites of the scaffold domain of RNase E can be crucial for efficient processing. In this respect phage φKZ and Dip act in a similar manner to phage T7 which phosphorylates the scaffold domain of RNase E to protect its transcripts from degradation by the host (*Marchand et al., 2001*).

Taking together our in vitro and in vivo results, it could be expected that the levels of RNA during a φKZ infection would be high, since both degradation of bacterial and phage RNA by the RNA degradosome are affected. Indeed, RNAseq data show that the total amount of cellular RNA extracted increased over five-fold during infection (*Ceyssens et al., 2014*). However, it was found that 35 min after the start of infection, 98.5% of the non-rRNA and non-tRNA could be mapped to the phage genome, suggesting a decrease of host RNA. Therefore, we hypothesize that at the start of infection by φKZ the bacterial RNA might be actively degraded. The mechanism for this is unclear, but could involve inhibition of the host RNA polymerase to impede nascent transcript genesis, and in parallel, the acceleration of host RNA decay by a yet uncharacterized enzyme of φKZ or a φKZ encoded activator of the hosts RNA degradation machinery. The production of the degradosome activating protein Srd by phage T4 has recently been reported (*Qi et al., 2015*), and although a homologue of Srd cannot be identified in the φKZ genome it is plausible that a functionally equivalent protein may be produced during the early stage of infection. Subsequently, Dip is produced to inhibit the activity of the RNA degradosome and protect the newly synthesized phage RNA. Since Western blot analyses indicate that Dip persists once it is produced, the stability of phage RNA can

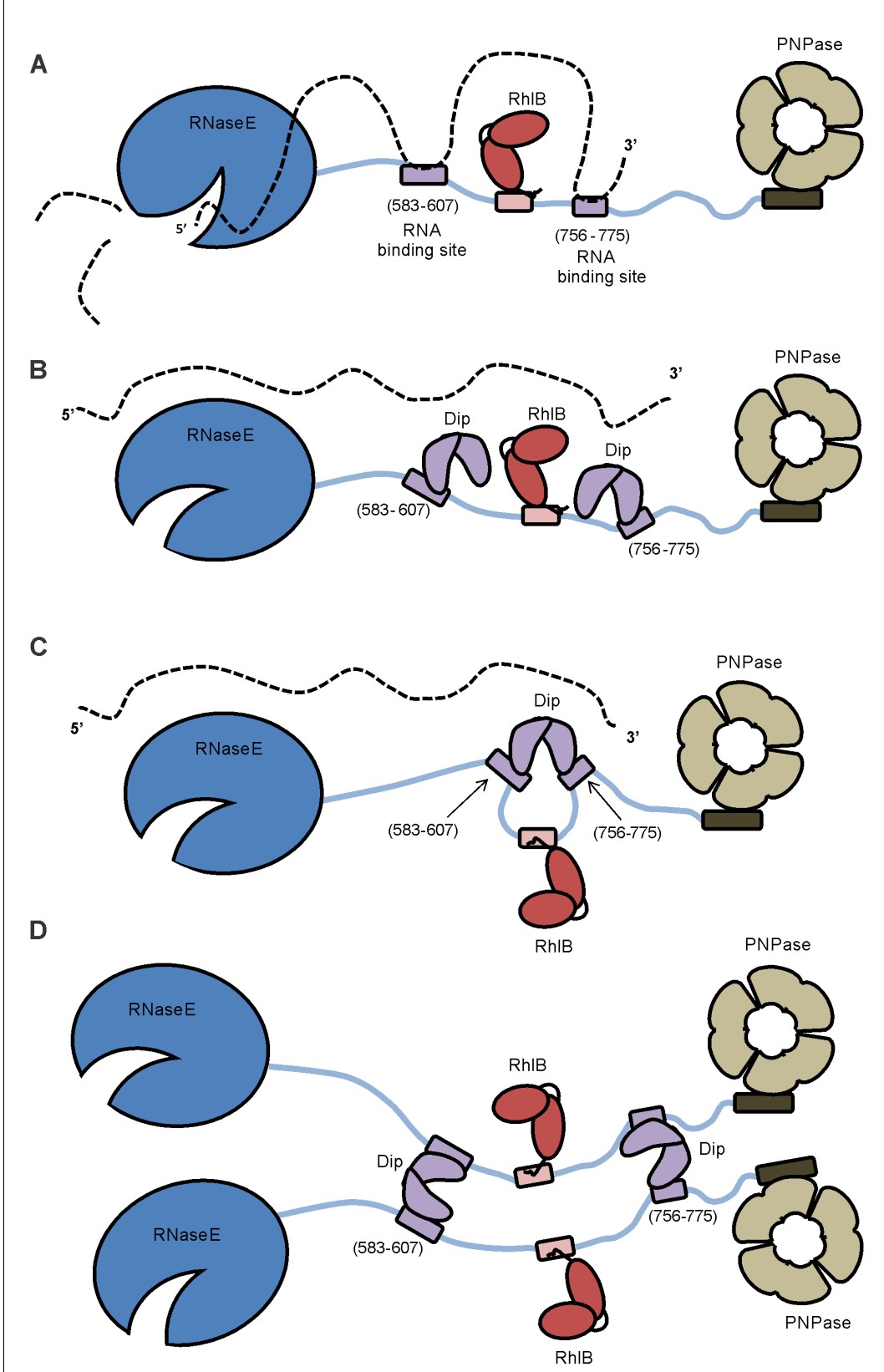

**Figure 8.** Model of the working mechanism of Dip. (**A**) The wild type *P. aeruginosa* degradsome consisting of the RNase E subunit (blue), the ATP-dependent RNA helicase RhlB (red) and the PNPase (beige). The RNA molecule (dotted line) binds to the RNA-binding sites of the scaffold domain of
*Figure 8 continued on next page*

*Figure 8 continued*

RNase E (purple) and is cut by the catalytic domain of RNase E. (**B**) A putative model hypothesizing that one Dip-dimer binds to each RNA-binding site during the infection by phage φKZ. (**C**) A second model hypothesizing that one Dip-dimer binds to both RNA-binding sites at the same time, yielding a looping of the scaffold domain of RNase E. (**D**) A model in which the Dip-dimers form a link between two RNA degradosome protomers.

be maintained during the entire infection cycle. It remains to be determined whether the production of Dip is essential for efficient infection by phage φKZ.

Dip is the third phage protein reported that has independently evolved to influence the activity of the RNA degradosome of its host, the others being gp0.7 of phage T7 which phosphorylates the scaffold domain of RNase E to protect its transcripts from degradation by the host, and Srd of phage T4 which enhances the activity of RNase E (*Marchand et al., 2001*; *Qi et al., 2015*). The function of these proteins opens the possibility that the modulation of RNA degradosome activity might be a recurrent strategy of bacteriophages to support efficient phage infection. The inhibition of bacterial host nucleases by phage encoded proteins may be a more common strategy than had previously been appreciated. Recently it has been shown that the CRISPR-Cas system of *P. aeruginosa* can be inhibited by several phage encoded proteins (*Bondy-denomy et al., 2015*), two of which act by binding and blocking DNA binding sites on the CRISPR-cas machinery, reminiscent of the mode of action of Dip. Moreover, the observation that Dip also has an inhibitory effect on the RNA degradosome of *E. coli*, suggests that the RNA binding sites within the CTD of RNase E are more conserved (either at sequence or structure level) than had previously been thought. Therefore, it can be speculated that there may exist functional homologues of Dip in other phages that target host machinery of RNA metabolism. Finally, the identification and biological understanding of the function of Dip opens the possibility of developing of new biotechnological tools to modulate RNA turn over and regulation in bacteria.

## Materials and methods

### Bacterial strains, phages and media

*P. aeruginosa* PAO1 was used during all manipulations (*Stover et al., 2000*). A *Strep*-tag II was fused to the C-terminus of RNase E (PA2976) by homologous recombination creating PAO1 *rne::StrepII* (*Lesic and Rahme, 2008*). Three *E. coli* strains were used: *E. coli* TOP10 (Life Technologies, Carlsbad, CA) for cloning procedures, *E. coli* BL21 (DE3) pLysS (Life Technologies) for heterologous expression of proteins and *E. coli* BTH101 (Euromedex, Souffelweyersheim, FR) for bacterial two-hybrid assays. Bacteria were grown in Lysogeny Broth (LB) (with appropriate antibiotics) at 37°C, unless stated elsewhere.

Seven *P. aeruginosa* specific phages were used: phage 14–1, φKZ, LUZ19, LKA1, LUZ24, PEV2 and YuA (*Van den Bossche et al., 2014*). The phages were stored in phage buffer (10 mM Tris-HCl pH 7.5, 10 mM $MgSO_4$, 150 mM NaCl).

### Affinity purifications and mass spectrometry

The engineered *rne::StrepII* strain was used for affinity purifications, after infection by one of the seven phages as described in *Van den Bossche et al. (2014)*. Eluted protein samples were loaded on a SDS-PAGE gel, after which the gel was cut into slices and subjected to a tryptic digest. ESI-MS/MS analyses were performed on a LCQ Classic (ThermoFinnigan, San Jose, CA, US) equipped with a nano-LC column switching system as described in *Dumont et al. (2004)*. The mass spectrometry proteomics data have been deposited to the ProteomeXchange Consortium (*Vizcaíno et al., 2014*) via the PRIDE partner repository with the dataset identifier PXD003285 and 10.6019/PXD003285.

### Protein expression and purification

The RNA degradosome of *P. aeruginosa* was purified as described in *Van den Bossche et al. (2014)*, using a 2 l culture of the PAO1 Rne::*StrepII* strain at $OD_{600\ nm}$ 0.6. The RNA degradosome of *E. coli* and fragment of the CTD were expressed and purified as described by *Tsai et al. (2012)*.

Dip and Dip-mutants were fused to a His-tag using the pEXP5-TOPO vector (Life Technologies) and transformed into *E. coli* BL21 (DE3) pLysS cells. The proteins were expressed and purified using a HisTrap HP column followed by size exclusion chromatography using a superdex 200 16/600 column (GE life sciences, Little Chalfont, UK) according the manufacturer's protocol. Purified proteins were either used directly for crystallization experiments or supplemented with 10% glycerol and stored at −80°C.

Fragments of the *P. aeruginosa* RNA degradosome were N-terminally fused to a GST-tag by cloning them into a pGEX-6P-1 vector using the BamHI and XhoI restriction sites (GE life sciences). The proteins were purified by Glutathione affinity chromatography followed by Heparin affinity chromatography (for RNase E constructs 583–607, 583–636, 756–775, 756–835 and 756–901) and finally size exclusion chromatography using a superdex 75 16/600 column (GE life sciences).

## Crystallography

A seleno-methionine derivative of Dip was produced using a metabolic inhibition method. His-tagged Dip was transformed into *E. coli* BL21 (DE3) pLysS cells grown in 1 L of M9 medium (42.2 mM $Na_2HPO_4$, 22 mM $KH_2PO_4$, 18.6 mM $NH_4Cl$, 8.5 mM NaCl, 1 mM $MgSO_4$, 0.4% (w/v) glucose, 0.00005% (w/v) vitamin B, 4.2 mg/ml $FeSO_4$)). At $OD_{600nm}$ 0.3, amino acids were added (final concentrations: 100 mg/l L-lysine, 100 mg/l L-phenylalanine, 100 mg/l L-threonine, 50 mg/l L-isoleucine, 50 mg/l L-leucine, 50 mg/l L-valine, 50 mg/l L-Seleno-methionine). 20 min after this addition, the production of Dip was induced with 1 mM IPTG. After 6 h, cells were harvested and proteins were purified as for wild type Dip. The best diffracting crystals of seleno-methionine Dip were produced in a condition of 100 mM $KH_2PO_4$, 100 mM $NaH_2PO_4$, 100 mM MES pH 6.0, 800 mM NaCl and 0.2 M sodium thiocyanate.

Data collected at beamline I24 of Diamond light source were used to solve the structure of Dip using the single wavelength anomalous diffraction method (SAD), utilizing the anomalous diffraction of the incorporated selenium atoms. An initial model of Dip was built automatically by the Phenix Autosolve pipeline (*Adams et al., 2010*), and the initial model was completed and improved with iterative cycles of refinement using Refmac5 (*Murshudov et al., 1997*) and manual model building in Coot (*Emsley and Lohkamp, 2010*).

Crystals of Dip in complex with the RNase $E_{756-775}$ were obtained by adding chemically synthesized peptide (Cambridge Research Biochemicals, Billingham UK) directly to Dip at a 1.5:1 molar ratio (peptide:Dip) prior to crystallisation. Hexagonal crystals of the Dip:RNaseE$_{756-775}$ complex were obtained in a condition of 100 mM $KH_2PO_4$, 100 mM $NaH_2PO_4$, 100 mM MES pH 6.0, 300 mM NaCl and 0.2 M sodium thiocyanate. Data collected at the beamline I04 of Diamond light source was used to solve the structure of Dip:RNaseE$_{756-775}$ by molecular replacement, and density for the RNase E peptide was apparent when processing the crystallographic data in the low symmetry P1 space group. Eight amino acids of the peptide were cautiously modelled into this density for each of the six protomers of Dip in the asymmetric unit. The crystal structures of Dip and Dip:RNaseE$_{756-775}$ are deposited at the protein data bank with accession codes 5FT0 and 5FT1.

## In vitro pull down and mass spectrometry

*P. aeruginosa* or *E. coli* cell lysate (collected at $OD_{600\ nm}$ 0.6) was prepared from cells resuspended in 20 ml of 'pull down' buffer (20 mM Tris pH 7.5, 200 mM NaCl, 20 mM imidazole) and cell lysis by three passages through a high-pressure homogenizer (Emulsiflex, Mannheim, DE). 0.5 mg of His-tagged Dip was added to a His-select nickel affinity spin column (Sigma Aldrich, St. Louis, MO, US), followed by 10 ml of the lysate, while an additional 10 ml of lysate was loaded on a blank spin column as a negative control. The spin columns were washed three times with 'pull down' buffer, and proteins were eluted in 50 µl 'pull down' buffer supplemented with 500 mM imidazole (15 min, 4°C). The eluted fraction was visualized by SDS-PAGE, and protein bands were identified by in gel mass spectrometry (PNAC facility, University of Cambridge).

## ELISA

ELISA was performed in Ni-NTA HisSorb Strips (Qiagen, Hilden, DE) according the manufacturer instructions. A 1:5,000 dilution of monoclonal anti-*Strep*-tag II antibodies conjugated to HRP (IBA, Goettingen, DE), the 1-Step Slow TMB-ELISA substrate and Stop Solution (Thermo Scientific,

Waltham, MA, US) were used for colorometric detection at $OD_{450nm}$ after 30 min of incubation. All reactions were performed in triplicate and wells without Dip, without RNA degradosome or without both proteins were used as a negative control.

## Mobility shift assays

A dilution of Dip was mixed with a dilution of an RNA degradosome fragment and/or an RNA fragment in 20 mM Tris pH 7.5 and 200 mM NaCl, and incubated for 10 min at $T_R$. After the addition of loading dye (0.2% (w/v) bromophenol blue, 300 mM DTT and 50% (w/v) glycerol), the samples were loaded on a 10% native polyacrylamide gel and run in running buffer (25 mM Tris, 250 mM glycine) in an electric field of 150 V. RNA was visualized by staining the gel with SYBR-gold stain, proteins by Coomassie staining.

## Bacterial two-hybrid

Bacterial two-hybrid assays were performed using the BACTH System kit (Bacterial adenylate cyclase two-hybrid system kit, Euromedex) (*Karimova et al., 1998*). Dip was cloned into the high copy number vectors fused to the N-terminal (pUT18) and C-terminal (pUT18C) end of the T18 domain of adenylate cyclase and into the low copy number vectors fused to the N-terminal (pN-25) and C-terminal (pKT25) end of the T25 domain (*Claessen et al., 2008*). The components/fragments of the RNA degradosome (*Figure 2A*) were cloned in pUT18C and pN-25. To screen for interactions each combination of phage and bacterial genes/fragments was co-transformed. Dilutions of an overnight culture were spotted on synthetic minimal M63 medium. β-galactosidase activity was measured quantitatively using a Miller assay (*Zhang and Bremer, 1995*).

## Western blot

Western blot was performed as described in *Van den Bossche et al. (2014)*. A 1/5000 dilution of polycolonal anti-Dip antibodies (produced in rabbits by Pharmlabs, KU Leuven, Belgium) were incubated with the membrane during one hour at room temperature, followed by a 1/5000 dilution of Anti-Rabbit IgG (H+L) antibodies conjugated by Horse Radish Peroxidase during one hour at room temperature. Detection was carried out by enhanced chemiluminescence.

## In vitro RNA degradation assay

A mixture of 0.05 µM RNA degradosome and 0.5 µM Dip in reaction buffer (25 mM Tris pH 8.0, 10 mM $MgCl_2$, 25 mM NaCl, 25 mM KCl, 1 mM DTT, 0.5U/ml of RNaseOUT) was incubated for 7 min at 37°C. Subsequently, the appropriate RNA fragment was added and incubated at 37°C. At certain time points, 10 µl aliquots were taken and the reaction was stopped by adding 10 µl proteinase K buffer (200 mM Tris pH 8.0, 25 mM EDTA, 300 mM NaCl, 2% (w/v) SDS, 0.5 mg/ml Tritirachium album Proteinase K) and incubation at 50°C for 30 min. 10 µl RNA loading dye was added and the samples were boiled for 5 min at 95°C. Samples were loaded on an 8% denaturing 7 M urea 19:1 acrylamide/bisacrylamide gel and run in TBE buffer for 1 hr in an electric field of 200 V. RNA was visualized with SYBR gold stain.

## qRT-PCR

Quantitative real-time PCR assays were carried out on a Rotor-Gene centrifugal real-time cycler (Qiagen) and analysis were performed on 3 µl of each sample in a total reaction volume of 15 µl using Absolute QPCR SYBR Green mix (Thermo Scientific) and 300 nM of both primers (Forward: GACG TACACGCGAAAGACCTG, Reverse: CTCGCCCAGAGCCATATTGTA). Thermocycles consisted of an initial 15 min denaturation at 95°C and 40 amplification cycles with an annealing temperature of 55.5°C according the manufacturer instructions. Quantification of the template was calculated from standard curves generated from a tenfold dilution series of genomic DNA, using the Rotor-Gene 6000 Series Software 1.7 (Qiagen). The data were normalized to the total RNA content of the samples and plotted in a semi-logaritmic graph. RNA half-lifes were calculated as the ln2/k, presuming first-order kinetics.

### Expression of phage proteins in *P. aeruginosa*

Dip was cloned into the pENTR/SD/D-TOPO vector (Invitrogen, Carlsbad, CA) and transferred to a pHERD20T-GW *E. coli* - *P. aeruginosa* shuttle vector using the Gateway cloning system (Invitrogen). Dip-E214A/E222A was cloned directly in pHERD20T. After transformation to *P. aeruginosa*, a dilution of an overnight culture was spotted on minimal medium ± 0% - 0.1% - 1% arabinose and incubated overnight at 37°C. The growth curves were monitored by a Bioscreen CTM spectrophotometer (Labsystems). Cells were visualised using a Nikon Eclipse Ti Time-Lapse Microscope (Nikon).

## Acknowledgements

We would like to thank Erik Royackers (Hasselt University, Belgium) for the technical support, the staff at Diamond Light source (Harwell, UK) for use of facilities and Boris Görke (University of Vienna, Austria) for providing the plasmids for the bacterial two-hybrid analyses. We acknowledge the PRIDE team for the deposition of our data to the ProteomeXchange Consortium. AV is doctoral fellow supported by the 'Fonds voor Wetenschappelijk Onderzoek' (FWO, Belgium). SH, KB and BFL are supported by the Wellcome Trust. This research was further supported by Grant G.0599.11 from the FWO, the SBO-project 100042 of the IWT ('Agentschap voor Innovatie door Wetenschap en Technologie in Vlaanderen), the JPN project R-3986 of the Herculesstichting and the grants CREA/09/017 and IDO/10/012 from the KU Leuven Research Fund.

## Additional information

### Funding

| Funder | Grant reference number | Author |
|---|---|---|
| Fonds Wetenschappelijk Onderzoek | G.0599.11 | An Van den Bossche Pieter-Jan Ceyssens Hanne Hendrix Rob Lavigne |
| Agentschap voor Innovatie door Wetenschap en Technologie | SBO 100042 | An Van den Bossche Pieter-Jan Ceyssens Rob Lavigne |
| Fonds Wetenschappelijk Onderzoek | Scholarship | An Van den Bossche |
| Wellcome Trust | scholarship | Steven W Hardwick Ben F Luisi |
| Onderzoeksraad, KU Leuven | GOA Bacteriophage Biosystems | Abram Aertsen Rob Lavigne Marc De Maeyer |
| Onderzoeksraad, KU Leuven | CREA/09/017 | Abram Aertsen Rob Lavigne |

The funders had no role in study design, data collection and interpretation, or the decision to submit the work for publication.

### Author contributions

AVdB, SWH, BFL, Conception and design, Acquisition of data, Analysis and interpretation of data, Drafting or revising the article; P-JC, RL, Conception and design, Analysis and interpretation of data, Drafting or revising the article; HH, MV, TD, KJB, MDM, J-PN, Acquisition of data, Analysis and interpretation of data; AA, Conception and design, Drafting or revising the article

### Author ORCIDs

Rob Lavigne, http://orcid.org/0000-0001-7377-1314

## Additional files

### Supplementary files

• Supplementary file 1. Mass spectrometry results for the Rne::StrepII affinity purification. Table 1. MS results of the affinity purifications on Rne::StrepII, infected with one of seven Pseudomonas phages. Table 2. Diffraction statistics and refinement statistics of the crystals of Dip.

### Major datasets

The following datasets were generated:

| Author(s) | Year | Dataset title | Dataset URL | Database, license, and accessibility information |
|---|---|---|---|---|
| An Van den Bossche, Jean-Paul Noben | 2015 | Mass spectrometry data set | https://www.ebi.ac.uk/pride/archive/projects/PXD003285 | Publicly available at the PRIDE Archive (accession no: PXD003285) |
| Steven W Hardwick | 2016 | Crystal structure of gp37(Dip) from bacteriophage phiKZ bound to RNase E of Pseudomonas aeruginosa | https://www.ebi.ac.uk/pdbe/entry/pdb/5ft1 | Publicly available at the RSCB Protein Data Bank (accession no. 5tf1) |
| Steven W Hardwick | 2016 | Crystal structure of gp37(Dip) from bacteriophage phiKZ | https://www.ebi.ac.uk/pdbe/entry/pdb/5ft0 | Publicly available at the RCSB Protein Data Bank (accession no. 5ft0) |

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
