## [Decision Letter]

Thank you for submitting your article "Structural elucidation of a novel mechanism for the bacteriophage-based inhibition of the RNA degradosome" for consideration by *eLife*. Your article has been reviewed by three peer reviewers, one of whom is a member of our Board of Reviewing Editors and the evaluation has been overseen by James Manley as the Senior Editor. The following individuals involved in review of your submission have agreed to reveal their identity: Udi Qimron (Reviewer #3).

The reviewers have discussed the reviews with one another and the Reviewing Editor has drafted this decision to help you prepare a revised submission.

Summary:

In this manuscript, Van den Bossche et al. report the discovery of a giant phage-encoded protein, gp37/Dip that inhibits host cell RNA degradation by associating with the two RNA binding sites of *Pseudomonas aeruginosa* RNase E. The study, which includes a 2.2Å structure of Dip, is very complete and reveals yet another interesting mechanism by which bacteriophage proteins can interfere with the host cell.

Essential revisions:

1) While the in vitro data are strong, the study would be more compelling with additional documentation that Dip affects RNA stability in vivo and/or further information about Dip activity throughout the infection cycle. Thus, the authors should carry out RNAseq normalized to the DNA content to show global RNA reduction following Dip expression. Carrying out such assays at several point during the infection cycle by a wt phage versus Dip deletion mutant would be a better than exogenous Dip expression from a plasmid.

2) One other experiment that could put the "icing on the cake" is to use the point mutants in Dip that reduce interaction with RNase E, and show that the mutant no longer exerts DIP's core functions. Such a mutation would serve as an additional and possibly informative control, and bring the story "full circle". If the authors have such data, if would be terrific to include.

---

## [Author Response]

*Essential revisions:*

*1) While the in vitro data are strong, the study would be more compelling with additional documentation that Dip affects RNA stability in vivo and/or further information about Dip activity throughout the infection cycle. Thus, the authors should carry out RNAseq normalized to the DNA content to show global RNA reduction following Dip expression. Carrying out such assays at several point during the infection cycle by a wt phage versus Dip deletion mutant would be a better than exogenous Dip expression from a plasmid.*

We agree that more information about Dip activity in vivo throughout the infection cycle would be desirable, and as such we have now raised an antibody specific to Dip and have included a Western blot figure to show the levels of Dip protein persist during the phage infection cycle (Figure 6). Carrying out RNAseq analyses during φKZ infection is practically not feasible due to the difficulties in normalizing results. As demonstrated by Ceyssens et al. 2014, in parallel to the decay of host RNA, host DNA is degraded during infection by φKZ so that it is not possible to normalize data to the DNA content. Also, experiments comparing wild-type phage to a Dip deletion mutant are not currently possible because there are no readily available approaches to genetically manipulate phiKZ phages, hence the requirement of expression of exogenous Dip from a plasmid. An antisense RNA expression inhibition approach proved unsuccessful.

*2) One other experiment that could put the "icing on the cake" is to use the point mutants in Dip that reduce interaction with RNase E, and show that the mutant no longer exerts DIP's core functions. Such a mutation would serve as an additional and possibly informative control, and bring the story "full circle". If the authors have such data, if would be terrific to include.*

This is a very good suggestion. We have now performed experiments to over-express the Dip mutants that are no longer capable of binding to RNase E in *P. aeruginosa*. Cells expressing the mutants no longer have the crescent-shape morphology that follows expression of the wild type protein. Images of these cells have been included in Figure 7—figure supplement 2 and are discussed in the Results section.